# Atherosclerosis and the Capillary Network; Pathophysiology and Potential Therapeutic Strategies

**DOI:** 10.3390/cells9010050

**Published:** 2019-12-24

**Authors:** Tilman Ziegler, Farah Abdel Rahman, Victoria Jurisch, Christian Kupatt

**Affiliations:** 1Klinik & Poliklinik für Innere Medizin I, Klinikum rechts der Isar, Technical University of Munich, 81675 Munich, Germany; tilman.ziegler@tum.de (T.Z.); farah.abdel-rahman@tum.de (F.A.R.); victoria.jurisch@googlemail.com (V.J.); 2DZHK (German Center for Cardiovascular Research), Partner Site Munich Heart Alliance, 80802 Munich, Germany

**Keywords:** atherosclerosis, pericyte, rAAV, capillary, endothelial cells

## Abstract

Atherosclerosis and associated ischemic organ dysfunction represent the number one cause of mortality worldwide. While the key drivers of atherosclerosis, arterial hypertension, hypercholesterolemia and diabetes mellitus, are well known disease entities and their contribution to the formation of atherosclerotic plaques are intensively studied and well understood, less effort is put on the effect of these disease states on microvascular structure an integrity. In this review we summarize the pathological changes occurring in the vascular system in response to prolonged exposure to these major risk factors, with a particular focus on the differences between these pathological alterations of the vessel wall in larger arteries as compared to the microcirculation. Furthermore, we intend to highlight potential therapeutic strategies to improve microvascular function during atherosclerotic vessel disease.

## 1. Introduction

Atherosclerosis remains an entity with continually growing incidence associated with a variety of disease states, which include ischemic stroke, peripheral artery disease and coronary artery disease. Taken together, cardiovascular disease, which can be seen as an expression of advanced atherosclerosis, account for 17.9 million deaths or 31% of all deaths per year globally, thus ranking it the number one cause of mortality today [1]. Multiple risk factors contributing to atherosclerosis have been identified and well-studied, mainly arterial hypertension, hypercholesterolemia, nicotine abuse and diabetes.

Atherosclerosis describes a pathological remodeling of the arterial wall initiated by the accumulation of lipids in the sub-endothelial layer of arteries. The retention of lipids triggers an inflammatory reaction leading to the invasion of multiple classes of leukocytes. This inflammatory state facilitates further endothelial dysfunction and remodeling of the extracellular matrix (ECM), ultimately leading to the formation of calcified, vulnerable plaques prone to rupture, which can lead to complete vessel occlusion via platelet activation and thrombosis.

This process of increasing vascular remodeling manifests initially as diffuse thickening of the Tunica intima, the innermost vascular layer, and an increase of intima thickness relative to the underlying Tunica media. Interestingly, these early stages in remodeling have been observed already at early ages, starting in the second decade of life [2]. The diffuse intimal thickening is mainly driven by the accumulation of lipids [3]. As proposed by Williams and Tabas in their response-to-retention hypothesis in 1995 [4,5], this accumulation of ECM-associated lipoproteins constitutes the initial step in the formation of an atherosclerotic lesion. The ECM-protein class of proteoglycans in particular has been identified as a key binding partner for lipid-complexes owing to their high affinity for lipoproteins [6,7]. Subsequently, the lipoprotein-proteoglycan complexes, which are prone to oxidation and aggregation, represent a source of oxidative stress on the surrounding endothelial cells and vascular smooth muscle cells (vSMC). This process induces the recruitment of macrophages, in part due to the increase in SMC-derived Monocyte chemoattractant protein-1 [8], which phagocytose the lipoprotein-proteoglycan complexes, leading to the accumulation of foam cells in the atherosclerotic plaque [9,10]. Owing to the increased number of macrophages and the subsequent release of inflammatory cytokines, vascular smooth muscle cells change from their resting state into a more fibroproliferative condition, e.g. they display a drastic increase in activation and expansion. They simultaneously demonstrate a heightened susceptibility to apoptosis, mediated by the induced expression of the pro-apoptotic regulator BAX (Bcl-2-associated X protein) [11]. In addition, activated vSMCs produce transforming growth factor β (TGF-β), tissue factor (TF) and further proteoglycans, thus attracting more lipoproteins and additional macrophages, which further worsens the progression of the atherosclerotic lesion [12,13,14]. In the later stages of atherosclerotic plaque formation, atherosclerotic lesions can transform into thin cap fibroatheromas, characterized by a thin fibrous cap containing calcifications covering a necrotic lipid core. These atherosclerotic lesions are prone to rupture, exposing the blood stream to the underlying extracellular matrix, containing Von Willebrand factor, collagen and fibrin. These ECM-proteins subsequently bind to platelets and lead to their activation and subsequent organization into a thrombus, resulting in the occlusion of the vessel [15,16,17].

In clinical practice, strategies for the treatment of atherosclerosis focus on the reduction of the risk factors of this pathological condition and on interventional or surgical revascularization. However, atherosclerosis is generally seen as a predominant problem of the macrocirculation with a focus on the formation of atherosclerotic plaques, rather than a disease affecting the whole circulatory system. In this review we discuss the impact of the known predominant risk factors—arterial hypertension, hypercholesterolemia and diabetes—on the development and progression of atherosclerosis with a focus on their influence on the microcirculation, e.g., arterioles, capillaries and venules. Furthermore, we highlight potential therapeutic strategies that might improve overall vascular function in atherosclerotic patients.

## 2. Cardiovascular Risk Factors Contributing to Atherosclerosis: The Macro

Arterial Hypertension represents a key driver in the development of atherosclerosis, and thus, cardiovascular disease. The prevalence of hypertension in ischemic stroke, coronary or peripheral artery disease lies reportedly around 60–90% depending on the localization of the atherosclerotic lesion [18,19]. Arterial hypertension can be divided into two classes: primary or essential hypertension, triggered by an interplay of underlying causes as well as secondary hypertension, caused by either endocrinological disorders or stenosis of the renal arteries. While the causes of arterial hypertension vary, the effect on the vascular system remains the same. It is currently unclear whether arterial hypertension represents the cause of vascular dysfunction or the result of it; however, a bidirectional interaction between hypertension and atherosclerosis appears the most likely explanation. Initial evidence that endothelial dysfunction causes hypertension stems from early observations that the inhibition of the endothelial nitric oxide synthase, which produces the potent vasodilator NO, leads to hypertension in human subjects [20]. One key regulator of arterial blood pressure is identified in the Renin-Angiotensin-Aldosterone system (RAAS), also regulating fluid and electrolyte homeostasis. Here, Angiotensin II, the main effector peptide of this system, has been demonstrated to directly induce endothelial dysfunction via the recruitment of macrophages to the vascular wall in a CCR2/MCP-1 dependent manner. Angiotensin II furthermore increases endothelial oxidative stress via NADPH oxidase–derived superoxide anion production, predominantly by interacting with the endothelial AT_1A_ receptor [21,22,23].

Hypercholesterolemia represents an additional risk factor with increasing prevalence in the development of cardiovascular disease [24,25]. Particularly, western style diets (high-fat and cholesterol, high-protein, high-sugar) lead to an increase in cholesterol, LDL-levels and LDL/HDL ratios [26]. Low density lipoproteins enter the vascular wall at predilection sites characterized by disturbed blood flow and preexisting endothelial dysfunction [27]. Once LDLs enter the vascular wall, they form complexes with proteoglycans (with versican, decorin, syndecan-4, biglycan and perlecan being the predominant proteoglycans in the vascular wall [28]) via the interaction of the LDL-component Apolipoprotein B [29]. This interaction facilitates changes to the lipid composition and the configuration of Apolipoprotein B [30], which enhances the oxidation of LDL to oxidized LDL (oxLDL) via reactive oxygen species generated by the activated endothelium and vascular smooth muscle cells [31]. This oxidation step represents a prerequisite for the detection and phagocytosis of LDL-particles by macrophages [32], leading to the formation of foam cells. This transformation increases the expression of inflammatory cytokines and oxidative stress markers in those macrophages [33]. Furthermore, oxLDL facilitates endothelial expression of leucocyte adhesion molecules (vacular cell adhesion protein 1, P-Selectin [34,35]) and cytokines [36], attracting additional macrophages, thus, enhancing the inflammatory state of the atherosclerotic lesion.

The last main contributor to the development of atherosclerosis can be identified in diabetes mellitus (DM). The hallmark feature of diabetes is the elevation of blood glucose levels (hyperglycemia). One key effect of hyperglycemia lies in the increased formation of superoxides, which enhances the oxidative stress of the vascular wall further. This process is partly mediated by the increased formation of advanced glycation end products (AGEs). Advanced glycation end products occur when excess glucose forms dicarbonyl compounds, which react spontaneously with amino groups of proteins [37]. These AGEs then bind to their respective receptor (RAGE), expressed on endothelial cells, macrophages and vascular smooth muscle cells. Particularly in endothelial cells, AGEs induce the activation of the NAD(P)H-oxidase [38] and also the expression of adhesion proteins and cytokines via the nuclear translocation of NFκB [39,40]. In macrophages, AGE-signaling enhances oxLDL uptake via an upregulation of CD36 and Macrophage Scavenger Receptor Class A [41]. Additionally, RAGE itself acts as an endothelial adhesion protein in concert with ICAM-1 [42], necessary for the adhesion of leukocytes. Further mechanisms enhancing ROS production upon hyperglycemic conditions are to be found in the Lipoxygenase pathway [43] and the Polypol pathway, respectively [44].

As discussed in the previous paragraphs, the formation of atherosclerotic lesions can be driven by multiple interdependent risk factors. However, arterial hypertension, diabetes mellitus and hypercholesterolemia also drastically change the functionality of the microcirculatory vessels, a fact rarely pointed out as compared to classical large-vessel pathologies. Therefore, in the following paragraphs we focus on the pathologies elicited by these risk factors in small vessels.

## 3. Hypertension, Hypercholesterolemia and Diabetes Mellitus in Capillaries: The Micro

Unlike larger vessels (arteries and veins), the smaller units of the vascular system lose their classical three layered structure consisting of Tunica intima, Tunica media and Tunica adventitia. While arterioles still contain a covering sheet of vascular smooth muscle cells throughout, capillaries are only sporadically covered in pericytes, a sort of mural cell closely related to vascular smooth muscle cells, but often lacking their contractile phenotype [45,46,47]. Venules on the other hand generally lack a complete cover of vSMCs. In their biological function, pericytes further differ from vascular smooth muscle cell layers due to their close interaction with endothelial cells. In the microcirculation, pericytes represent key regulators of endothelial quiescence, predominantly by secreting the growth factor Angiopoietin-1, which binds to the endothelial Tie-2 receptor [48]. Activation of this receptor tyrosine kinase facilitates the expression of survival factors, such as Survivin, and suppressing the expression of pro-apoptotic signaling molecules, like procaspase-9 and BAD (BCL2 Associated Agonist Of Cell Death) in endothelial cells [49,50]. Furthermore, Angiopoietin-1 enhances the recruitment of additional pericytes to the endothelial monolayer in a HB-EGF (heparin binding EGF like growth factor) and HGF (hepatocyte growth factor) dependent positive feedback loop [51,52]. In addition to Ang-1, pericytes regulate endothelial proliferation rates via TGF-β (transforming growth factor β) [53,54], bFGF (basic fibroblast growth factor) [55] and VEGF (vascular endothelial growth factor) [56]. Of note, pathological stimuli such as inflammation, hypoxia and neoplasia generally do not manifest themselves in a proliferation of mural cells in the microcirculation but, rather, a decrease in endothelial pericyte coverage. While this loss in pericytes has been widely reported, their fate remains unknown. Speculations range from de-differentiation to migration or apoptosis, depending on the particular vascular bed and stimulus [57,58,59,60]. Furthermore, the influx of macrophages into the vessel wall and subsequent transition into foam cells, one key driver in the development of atherosclerosis, does not occur in the same way in capillaries, since there is no comparable structure – e.g., tunica media – in these smallest vessels. These differences in morphology and in the reaction of pericytes to pathological stimuli also have an impact on the functional and morphological changes of capillaries exposed to those stimuli. However, while some reactions of the vascular system to the exposure to risk factors still remain in effect in capillaries (increased expression of endothelial adhesion molecules and of reactive oxygen species leading to a state of endothelial dysfunction), the following paragraphs will focus on the specific differences between the microcirculation and larger vessels in response to pro-atherosclerotic stimuli.

Arterial hypertension leads to an increase in the vascular pericyte coverage. Apart from this enhancement of the number of pericytes, this cell type also undergoes a transformation into a more vascular smooth muscle cell like phenotype, indicated by an increase in the expression of contractile proteins [61,62]. This effect is in part mediated by an upregulation of the endothelial-derived growth factor FGF-2 and interleukin-6 [63]. Interestingly, this increase is not accompanied by a gain in capillary density, but rather the opposite. Capillary rarefication is routinely seen both in human as well as in animal studies during arterial hypertension [64,65]. The growing number of pericytes and endothelial coverage with pericytes during arterial hypertension however appears to be unique to the hypertensive stimulus, since hypercholesterolemia, hyperglycemia and the subsequent inflammation elicited generally leads to a drastic decrease in pericyte coverage. To this end, hypercholesterolemia leads to a decrease in endothelial pericyte coverage in part via the downregulation of the endothelial NO synthase, a potent driver of microvascular mural cell recruitment [66,67,68]. In addition, hypercholesterolemia facilitates the above mentioned accumulation of reactive oxygen species and the increased recruitment of leukocytes [69]. Other effects seen during states of increased lipid deposition are the reduction of angiogenic sprouting via a downregulation of vascular endothelial growth factor (VEGF-A) and an additional reduction of endothelial N-Cadherin, the key anchoring protein with which endothelial cells and pericytes interact [70,71]. Diabetes as well seems to be associated with a drastic loss in pericytes. In this context, pericyte loss represents a key feature in the case of diabetic retinopathy. This complication of end-stage diabetes constitutes a well-studied phenomenon, due to the accessibility of the vascular bed to investigation both in human specimens as well as animal models. In both, capillary rarefication as well as a loss in pericytes is observed during diabetic retinopathy [72,73], while this process has also been demonstrated in additional vascular beds [74], mediated in part by advanced glycation end product accumulation [75].

## 4. Current and Future Treatment Strategies

Multiple therapies have been in clinical use to treat atherosclerotic lesions and the underlying cardiovascular risk factors. Two treatment options can be destinguished: firstly, the treatment of the predisposing factors causing atherosclerosis in order to reduce the progression of the disease once diagnosed. For primary arterial hypertension a host of antihypertensive drugs are available with varying efficacies and substance-class specific secondary effects. Notably, ACE-inhibitors and angiotensin 2 receptor antagonists not only lower the blood pressure but also reduce the degree of endothelial dysfunction by reducing leukocyte recruitment and the production of reactive oxygen species [76], an effect also demonstrated for calcium channel blockers [77]. Antidiabetic drugs too display pleiotropic effects beneficial for cardiovascular mortality. To this end, metformin and the novel class of PCSK9 inhibitors additionally reduce reactive oxygen species production [78,79]. Lastly, the pleiotropic effects of statins, the first line treatment option for hypercholesterolemia, have been well documented throughout the last decades. These drugs reduce endothelial cytokine production [80], production of reactive oxygen species [81] and vascular smooth muscle cell proliferation [82].

Mechanical revascularization, either via bypass surgery or percutaneous angioplasty, represents the second category in the treatment of atherosclerosis. These methods have demonstrated their merit over time for both coronary as well as peripheral artery disease. However, while extensive research has been undertaken to optimize surgery procedures and percutaneous vascular intervention strategies, no-reflow phenomena routinely occur in patients undergoing revascularization both in acute ischemic events as well as chronic vascular occlusion with rates varying from 2% up to 25%, depending on the vascular bed and the abruptness of occlusion [83,84]. In the case of acute myocardial or limb ischemia, these events can be attributed to a high thrombotic burden in the occluded vessel. However, the predominant vascular risk factors for the development of atherosclerotic lesions can have additional effects in the microcirculation as described above. In particular, the rarefication of capillaries and the dysfunctionality of the remaining capillaries, leading to a dysfunctional downstream vessels system, resulting in a drastic decrease in overall capillary diameter. This reduction in available runoff contributes to low-flow phenomena through recently implanted stents and increase the risk of stent thrombosis (see Figure 1).

Consequently, therapeutic strategies to ameliorate capillary rarefication and improve the functionality of the downstream capillary network need to be established. Since capillary stability crucially relies on proper pericyte adhesion to the endothelial tube, factors increasing pericyte abundance, as well as promoting angiogenic sprouting appear to be among the most promising targets to reduce the loss in capillaries during chronic.

One such agent can be found in the small peptide Thymosin β4. Thymosin β4 was first identified as an actin sequestering peptide binding G-actin in competition with Myocardin-related transcription factor A (MRTF-A) [85]. MRTF-A in its unbound form, i.e., free of G-actin binding, is capable to translocate into the nucleus, where it regulates the expression of SRF (serum response factor) target genes, in particular *CCN1* and *CCN2* [86]. CCN1 and CCN2 have been shown to promote angiogenic sprouting and vascular maturation via the recruitment of pericytes to newly formed vessels [87,88]. Increasing either the availability of Thymosin β4 or MRTF-A can promote MRTF-A nuclear translocation during chronic ischemia, as demonstrated in mouse and pig models of chronic limb ischemia as well as myocardial ischemia and reperfusion and hibernating myocardium [74,86,89]. Interestingly, Thymosin β4 has proven successful in improving myocardial perfusion in a model of chronic myocardial ischemia in otherwise healthy pigs and also hypercholesterolemic pigs and transgenic diabetic pigs [74,89], indicating its potential in the treatment of patients with underlying risk factors. Since arterial hypertension, hypercholesterolemia and diabetes mellitus are chronic disease states, a long term treatment seems preferable under these conditions. Thus, recombinant adeno-associated viral vectors (rAAV), as used in these studies, appear a favorable option, given their ability to facilitate long-term transgene expression with minimal genomic integration and low levels of host immune responses [90,91]. Another key regulator of angiogenic sprouting ameliorating capillary rarefication can be identified in vascular endothelial growth factor A (VEGF-A). VEGF-A promotes endothelial proliferation and tip-cell formation. However, long-term overexpression of VEGF-A leads to the formation of hemangioma-like structures with poor perfusion [92]. This effect can be prevented, when VEGF-A is administered as a combination treatment with a pericyte recruiting agent, such as PDGF-B, which stabilize newly formed vessels via the integration of pericytes into the sprouting vascular network [93]. Thus, a cotransfection of rabbits undergoing chronic hind limb ischemia with both rAAV.PDGF-B as well as rAAV. VEGF-A can induce collateral growth, increases capillary density and enhances the perfusion of the chronically occluded hind limb. Seeing as increased VEGF-A levels over prolonged periods of time lead to the formation of dysfunctional vessel, other modes of delivery might be advantageous. Overexpression of target genes in short bursts can be achieved via the transfection of modified RNA, which contains alternative nucleotides (pseudouridine, methylpseudouridine or 5-methyl-cytosine) to prevent TLR7/8 mediated host immune responses [94]. Using modRNA encoding for VEGF-A in mouse and pig models of chronic coronary occlusion, Carlsson et al. demonstrated a robust and short term VEGF-A expression, leading to an increase in both capillary and arteriole density. After proving the efficacy of an intramyocardial injection of VEGF-A modRNA, Gan et al. demonstrated in a recent phase Ia/b clinical study in diabetic patients, that localized injection of VEGF-A modRNA leads to a robust short term transgene expression without the induction of a significant immunresponse while locally improving perfusion [95]. Thus, capillary rarefication and loss of pericytes are treatment targets accessible for gene therapy approaches in vascular disease. However, other disease states accompanied by capillary rarefication might profit from similar gene therapy approaches, such as Duchenne muscular dystrophy (DMD). DMD is caused by mutations in the dystrophin gene, leading to the production of unstable, truncated and dysfunctional proteins [96,97]. While Duchenne muscular dystrophy is mostly recognized as a disease of the peripheral muscle and myocardium, it is also accompanied by capillary rarefication. This process contributes to the dire health status of patients afflicted by this genetic disorder by aggravating tissue ischemia [98]. In this context, our group was able to demonstrate that pigs lacking the exon 52 of the dystrophin gene (DMDΔ52), which leads to the expression of a similarly shortened and unstable dystrophin protein [99], also display a decrease in tissue capillary density and pericyte coverage. Once treated with rAAV containing a split Cas9 protein and guide RNAs targeting Exon 51, these animals not only regained expression of a functional dystrophin gene, but also showed a drastic improvement of tissue capillarization, a process accompanied by a reduction in CD68 positive macrophages (Figure 2, Moretti et al., Nature Medicine, accepted for publishing).

Interestingly, these findings combined highlight potential therapeutic strategies to target the rather neglected pathophysiological changes mediated by atherosclerotic risk factors in the microcirculation and represent a potential addition to the classical treatment strategies to ameliorate the disease burden of vascular occlusive disease.

## 5. Conclusions

Atherosclerosis represents a multifactorial disease mainly driven by arterial hypertension, hypercholesterolemia and diabetes mellitus, which, through the stenosis and occlusion of arteries, leads to organ ischemia and thus constitutes a main driver of mortality worldwide. The three main contributors to the development of atherosclerosis can originate from similar sources, such as sedentary lifestyle, western diet and obesity and exerted damage to the vessel wall via distinct but overlapping pathomechanisms. The hallmark of vascular alterations elicited by all three disease entities lies in the endothelial dysfunction seen in atherosclerosis, which is largely driven by an increase in endothelial activation with an elevated uptake of lipids, namely low-density lipoproteins, into the vascular wall. This process triggers the production of reactive oxygen species as well as the attraction of macrophages to the site of plaque formation, leading to their transformation into foam cells. All of those pathological alterations can enter into a positive feedback loop aggravating the development of plaque formation. One additional component in this disease progression is the proliferation of vascular smooth muscle cells, which can participate in the uptake of oxidized LDL and can also transform into foam cells, which serve as a source for inflammatory cytokines, attracting more macrophages. Herein lies one key difference between the macrocirculation (e.g., larger arteries) and the microcirculation exemplified by capillaries. Capillaries are surrounded not by vascular smooth muscle cells but by pericytes, a cell type related to vascular smooth muscle cells but with distinct functions in the vascular unit. Unlike vSMCs, pericytes react to the pathological stimuli elicited by hyperglycemia and hypercholesterolemia with a detachment from the underlying endothelium, resulting in further endothelial activation and apoptosis. This leads to capillary rarefication and reduced blood flow due to a decrease in capillary surface area.

Here, we highlighted potential therapeutic targets to improve microvascular dysfunction, namely by expressing proangiogenic growth factors and pericyte chemoattractants, either combined in one signaling molecule (as is the case for Thymosin β4) or in a cooperative fashion (such as the combined overexpression of VEGF-A and PDGF-B), all of which are mediated by a recombinant adeno-associated viral vector mediated overexpression, or the short-term burst expression of VEGF-A alone in the form of VEGF-A encoding modified RNA.

Taken together, the microcirculatory changes during atherosclerosis warrant further investigation and represent a worthwhile topic for additional studies.

## Figures and Tables

**Figure 1 cells-09-00050-f001:**
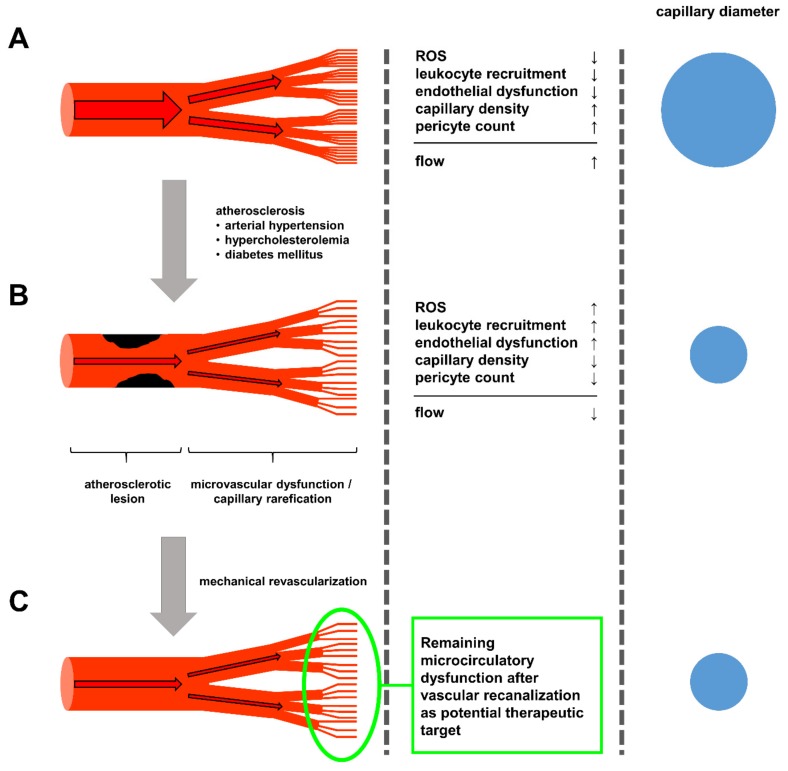
(**A**) The healthy circulatory system is characterized by minimal lipid accumulation in larger arteries and an overall low state of endothelial activation, leading to low levels of ROS production and leukocyte recruitment. (**B**) Upon prolonged exposure to the atherosclerotic risk factors arterial hypertension, hypercholesterolemia and diabetes, endothelial cells experience constant activation enhancing leukocyte recruitment, oxidative stress and loss of pericytes in the microcirculation, leading to capillary rarefication, limiting the potential blood flow through the now sparse capillary network. (**C**) Even after mechanical revascularization, via bypass operations or percutaneous angioplasty, the capillary rarefication remains, continuously limiting blood flow, thus hindering the recovery of the ischemic tissue and leaving newly opened vessels susceptible for restenosis and stent thrombosis. Here, strategies to improve capillary density, and thus, microcirculatory flow, appear to be worthwhile therapeutic targets in the treatment of atherosclerosis currently not yet addresses.

**Figure 2 cells-09-00050-f002:**
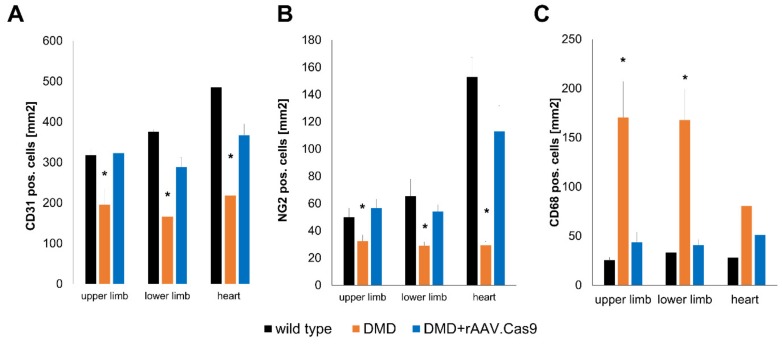
Effect of in vivo genome editing via rAAV and Cas9 mediated deletion of exon 51 of Duchenne muscular dystrophy on the vascularization and macrophage recruitment in the heart and upper and lower hind limb in DMDΔ52 pigs. (**A**) staining for CD31 positive endothelial cells highlights a significant decrease in capillary density in pigs suffering from Duchenne muscular dystrophy which ins ameliorated in pigs receiving Cas9 mediated Exon 51 deletion thus restoring dystrophin expression. (**B**) Similarly, edited DMD pigs display an amelioration of pericyte loss seen in dystrophin deficient pigs. (**C**) Lastly, the reduction in both endothelial cells as well as pericytes in dystrophin deficient pigs is accompanied by an increased recruitment of CD68 positive macrophages into the tissue, similarly to the recruitment seen in atherosclerotic states, which is again reversed upon normalization of dystrophin expression (**p* < 0.05 versus wild type and DMD+rAAV.Cas9, error bars are given as SEM).

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
