# Peer review of "Atherosclerosis and the Capillary Network; Pathophysiology and Potential Therapeutic Strategies"

_cells, 2019, doi:10.3390/cells9010050_

Round 1

Reviewer 1 Report

In the present paper a review of atherosclerosis risk factors with a particular focus of their effects on the microcirculation is made.

Moreover, a highlight on the potential therapeutic strategies that might improve overall vascular function in atherosclerotic patients is made.

The topic is original and interesting, possibly leading to new strathegies in future.

The manuscript is very well written and the topic is extremely well developed.

I would suggest to simplify paragraph 2 and 3 which appears to be clear and complete, but a little too verbose. 

Author Response

Reviewer 1

In the present paper a review of atherosclerosis risk factors with a particular focus of their effects on the microcirculation is made.

Moreover, a highlight on the potential therapeutic strategies that might improve overall vascular function in atherosclerotic patients is made.

The topic is original and interesting, possibly leading to new strathegies in future.

The manuscript is very well written and the topic is extremely well developed.

I would suggest to simplify paragraph 2 and 3 which appears to be clear and complete, but a little too verbose. 

We thank the reviewer for her/his comments. Paragraphs 2 and 3 have been simplified to enhance comprehensibility. Changes made are highlighted in red.

Reviewer 2 Report

In general, the present review is good overview of present literature concerning present knowledge of tissue microcirculation in atherosclerosis. The topic is relevant and not well covered in the present literature. The major concern is somehow long and complicated sentences (pages 1-4). Revision of these would make this interesting review easier to read and understand.

Comments:

P1 lines 16-18. The sentence is not logic. Pathophysiological changes during atherosclerotic plaque formation and microvascular system... something is lacking from the sentence?

P1 Line 22 lading or leading?

P1 lines 30-33 could this long sentence be re-written into shorter sentences to make it easier to read?

P2 lines 61-63 the sentence is somehow not easy to interpret.

P2 lines 63-66 long sentence with word while repeated several times. Could be re-written more compact sentences?

P4 lines 152-155 sentence requires revision

P4 lines 170-173 would benefit from revision

Author Response

Reviewer 2

In general, the present review is good overview of present literature concerning present knowledge of tissue microcirculation in atherosclerosis. The topic is relevant and not well covered in the present literature. The major concern is somehow long and complicated sentences (pages 1-4). Revision of these would make this interesting review easier to read and understand.

We thank the reviewer for her/his comments.

Comments:

P1 lines 16-18. The sentence is not logic. Pathophysiological changes during atherosclerotic plaque formation and microvascular system... something is lacking from the sentence?

P1 Line 22 lading or leading?

P1 lines 30-33 could this long sentence be re-written into shorter sentences to make it easier to read?

P2 lines 61-63 the sentence is somehow not easy to interpret.

P2 lines 63-66 long sentence with word while repeated several times. Could be re-written more compact sentences?

P4 lines 152-155 sentence requires revision

P4 lines 170-173 would benefit from revision

The specific sentences mentioned by the Reviewer have been rewritten. Furthermore, the complete manuscript has been carefully revised to enhance intelligibility. Changes have been marked in red.